# The structure of a 15-stranded actin-like filament from *Clostridium botulinum*

Fujiet Koh[1,2], Akihiro Narita[3], Lin Jie Lee[1,2], Kotaro Tanaka[3,4], Yong Zi Tan [5,6], Venkata P. Dandey[6], David Popp[1,7] & Robert C. Robinson[1,8,9]

Microfilaments (actin) and microtubules represent the extremes in eukaryotic cytoskeleton cross-sectional dimensions, raising the question of whether filament architectures are limited by protein fold. Here, we report the cryoelectron microscopy structure of a complex filament formed from 15 protofilaments of an actin-like protein. This actin-like ParM is encoded on the large pCBH *Clostridium botulinum* plasmid. In cross-section, the ~26 nm diameter filament comprises a central helical protofilament surrounded by intermediate and outer layers of six and eight twisted protofilaments, respectively. Alternating polarity of the layers allows for similar lateral contacts between each layer. This filament design is stiffer than the actin filament, and has likely been selected for during evolution to move large cargos. The comparable sizes of microtubule and pCBH ParM filaments indicate that larger filament architectures are not limited by the protomer fold. Instead, function appears to have been the evolutionary driving force to produce broad, complex filaments.

[1] Institute of Molecular and Cell Biology, A*STAR (Agency for Science, Technology and Research), Biopolis, Singapore 138673, Singapore. [2] Graduate School for Integrative Sciences and Engineering, National University of Singapore, Singapore 138632, Singapore. [3] Structural Biology Research Center, Graduate School of Science, Nagoya University, Furo-cho, Chikusa-ku, Nagoya 464-8601, Japan. [4] Department of Physics and Information Technology, Graduate School of Computer Science and Systems Engineering, Kyushu Institute of Technology, Kawazu 680-4, Iizuka, Fukuoka 820-8502, Japan. [5] Department of Biochemistry and Molecular Biophysics, Columbia University, New York, NY 10032, USA. [6] National Resource for Automated Molecular Microscopy, Simons Electron Microscopy Center, New York Structural Biology Center, New York, NY 10027, USA. [7] Fly&Fish Lab, Ebino, Miyazaki 889-4163, Japan. [8] School of Biomolecular Science and Engineering (BSE), Vidyasirimedhi Institute of Science and Technology (VISTEC), Rayong 21210, Thailand. [9] Research Institute for Interdisciplinary Science, Okayama University, Okayama 700-8530, Japan. Correspondence and requests for materials should be addressed to A. N. (email: narita.akihiro@f.mbox.nagoya-u.ac.jp) or to D.P. (email: thebamboorodmaker@gmail.com)

large DNA plasmids often encode segregation systems that ensure their faithful inheritance in bacteria. Functional ParMRC plasmid segregation systems comprise an actin-like polymerizing motor (ParM), a centromere-like specific DNA sequence (*parC*), and an adaptor protein (ParR), which attaches a plasmid, via the *parC*, to each end of the growing ParM filament or filament bundle[1,2]. Thus, the force generated from ParM polymerization is harnessed for plasmid segregation. Recently, structural studies have highlighted that ParMRC systems are highly plasmid specific. In particular, large variations in ParM filament geometries have been described that have differences in polarity, twist and the numbers of associating pairs of filament strands[3–5]. This variety of filament structures is likely required in cells that simultaneously maintain multiple plasmid types, where each unique ParM actin-like motor is dedicated to the survival of a single type, or small number of plasmid types. *Clostridium botulinum* often supports more than one plasmid, many of which are large and encode for neurotoxins[6,7]. Here we investigated the ParMRC cassette from pCBH, a 257 kb plasmid that carries the botulinum neurotoxin type B. This ParMRC cassette is also found on other *C. botulinum* plasmids, such as pCLK (267 kb) and pRSJ2_3 (245 kb), which encode neurotoxin types A and F, respectively.

## Results

**The pCBH ParMRC cassette**. In order to determine that the putative pCBH ParMRC cassette encodes functional elements (Fig. 1a), we determined that pCBH ParM quickly assembled on addition of ATP monitored by light scattering (Fig. 1b). Phosphate release, following nucleotide hydrolysis, was measured to have delayed kinetics (Fig. 1b), and pCBH ParM disassembly was

substantively slower, as observed by a gradual loss in light scattering (Fig. 1b). The critical concentration for assembly was estimated to be around 3 μM from the plot of maximum intensity values of light-scattering curves at different pCBH ParM concentrations (Fig. 1c). This compares with a similar value of 1.5–2 μM determined in vitro for the *Escherichia coli* R1 plasmid ParM[8], for which the cellular concentration of ParM has been estimated to be 12–14 μM[9]. Thus, the filament assembly parameters are in line with this well-characterized segregation system. Titration of DNA fragments generated via PCR from pCBH *parC* with increasing levels of pCBH ParR resulted in a defined mobility shift to larger molecular size, consistent with a specific interaction between pCBH ParR and pCBH *parC* (Fig. 1d). Together these interactions are consistent with the identification of the pCBH ParMRC cassette as a plasmid segregation system, since the ParM polymerizes and the ParR is able to interact with *parC*.

**CryoEM of the pCBH ParM filament**. Electron microscopy (EM) of negatively stained specimens and subsequently cryoEM images indicated that the pCBH ParM filaments are substantially thicker and straighter than F-actin[10] (Fig. 2). Estimation of the persistence length of the pCBH ParM filaments from the cryoEM images is 35 μm, which compares to 11 μm for the actin filament by the same method, consistent with previous reports (10–11 μm)[11,12]. These estimations will be dependent on solution conditions, nucleotide state, and the thickness of the ice, however they indicate that the pCBH ParM filaments are substantially stiffer than actin. The pCBH ParM filaments could be imaged under a wide range of conditions including high physiological salt concentrations typically found in bacterial cells. The condition used

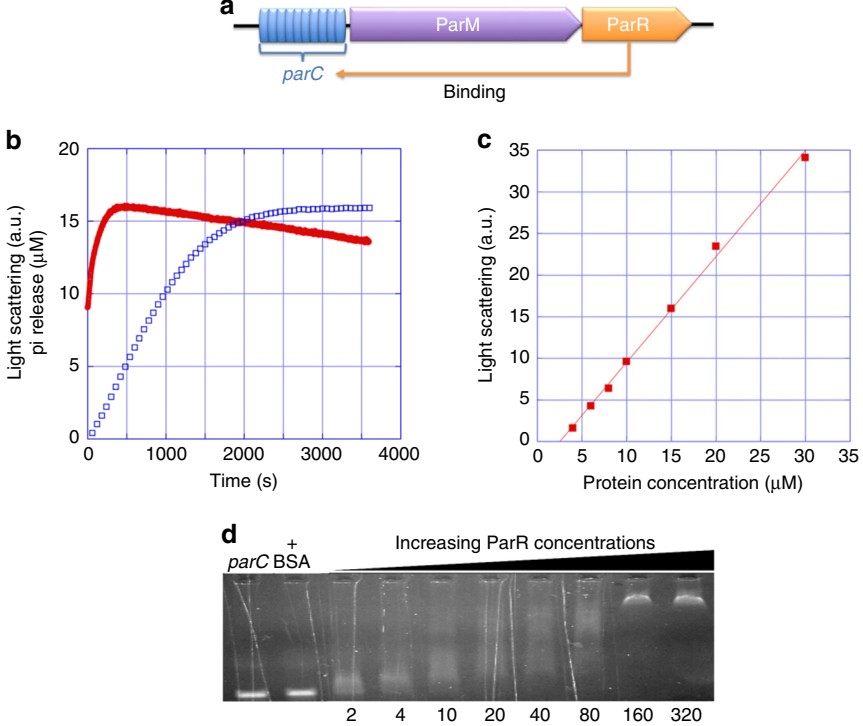

**Fig. 1** Interactions of the putative pCBH ParMRC cassette components. **a** Schematic of the operon. The complete sequence of *Clostridium botulinum* Prevot_594 plasmid pCBH (GenBank: CP006901.1) comprises *parC* (9901–10030), ParM (10031–11083; AJD29063.1), and ParR (11558–11935; AJD29378.1). **b** Typical light scattering curve of pCBH ParM polymerization (red, 15 μM) initiated by 2 mM ATP. Corresponding Pi release curve (blue). The Pi release rate was estimated from the linear slope to be ~10 nM/s. **c** Plot of the maximum light scattering intensity at different concentrations of pCBH ParM. The intersection of the maximum light scattering intensity vs the protein concentration on the *x*-axis gives an approximate critical concentration of ~3 μM. **d** EMSA of pCBH *parC* (20 nM) with increasing ratios of pCBH ParR indicated in μM

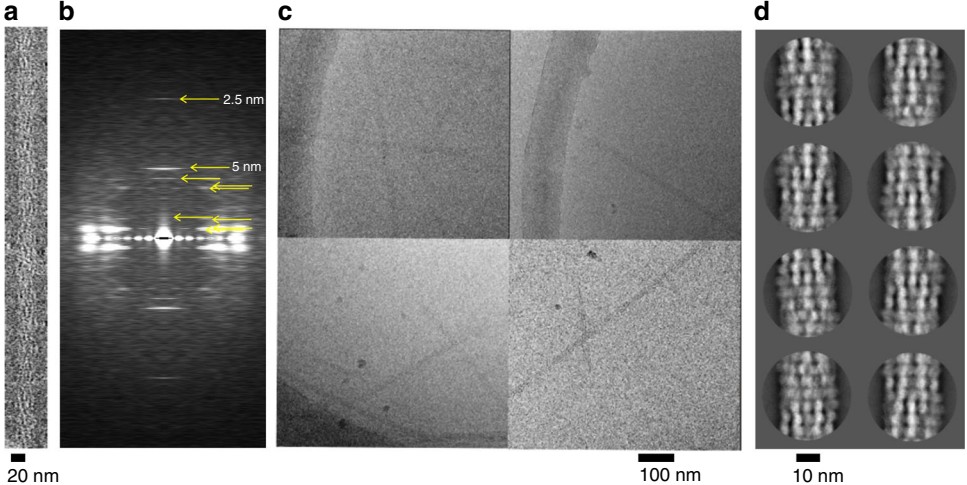

**Fig. 2** EM analysis of pCBH ParM filaments. **a** EM image of a negatively stained pCBH ParM filament. **b** Averaged layer lines from 50 negatively stained filament images. Prominent layer lines are indicated by yellow arrows **c** CryoEM images of pCBH ParM filaments. **d** Examples of 2D class averages. The eight largest classes are presented

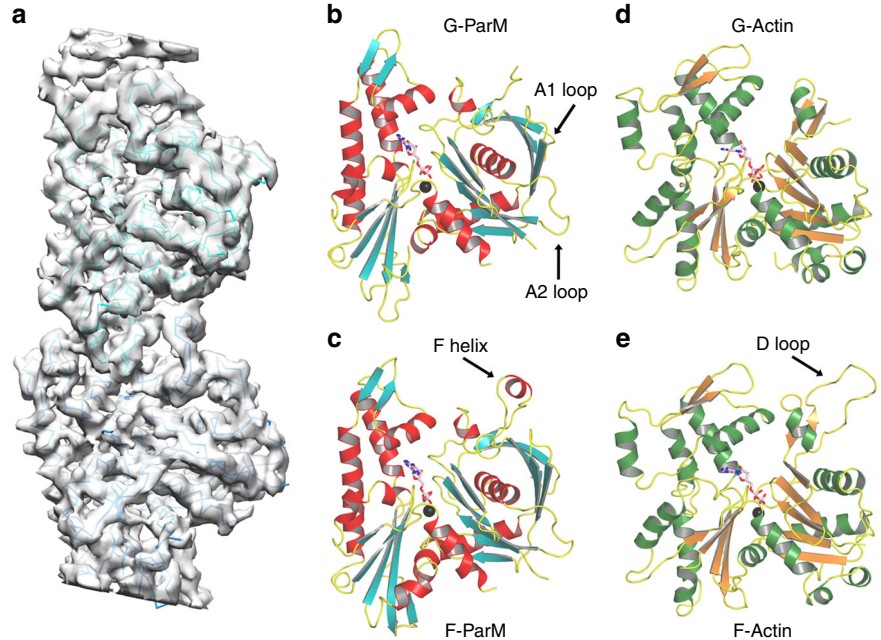

**Fig. 3** pCBH ParM protomer structures. **a** Averaged cryoEM density of the protofilament in the intermediate layer at 4.2 Å resolution with fitted model. **b** The pCBH ParM monomer crystal structure bound to $Mg^{2+}$-ADP. **c** The pCBH ParM cryoEM filament protomer structure bound to $Mg^{2+}$-ADP. **d** G-actin crystal structure bound to $Ca^{2+}$-ATP[16]. **e** F-actin cryoEM protomer structure bound to $Mg^{2+}$-ADP[15]. F helix, D loop, A1 loop, and A2 loop indicate regions with roles in filament formation. Data collection and refinement statistics are found in Supplementary Tables 1 and 2

to form the most homogeneous population for cryoEM imaging was 70 mM KCl, 7 mM $MgCl_2$, 2 mM ATP, 10 mM HEPES, pH 7.5. All filaments showed similar widths on the micrographs. We extracted 36,292 particles and selected 33,356 particles using Class2D in Relion[13,14], indicating more than 90% of the particles are homogeneous. The 2D class averages indicated a complex filament architecture (Fig. 2d), as did the averaged Fourier transform calculated from 50 negatively stained filament images (Fig. 2b). Due to this complexity, the helical parameters were determined by cryoelectron tomography (Supplementary Fig. 1, axial rise 5.2 nm, twist −50.1°). These parameters refined to a distance 5.03 nm and twist −50.4° with the cryoEM data. Helical averaging of the cryoEM density, from each cross-section of the

filament, based on these parameters led to a 4.7 Å map for the entire filament (Supplementary Fig. 2a, b). Within each cross-section, an intermediate layer consisting of six hexagonal proto-mers showed the best local resolution. Inter-strand averaging for this intermediate layer led to a 4.2 Å map for the protofilament (Fig. 3a and Supplementary Fig. 2).

**pCBH ParM protomer structure.** Using the *E. coli* R1 ParM cryoEM protofilament structure as a template, we designed a quadruple mutant of pCBH ParM that was predicted to prevent polymerization (Supplementary Fig. 3a). Mutations to subdomain 2 (F42D, I46D) and subdomain 3 (S298D and R299D) were

expected to disrupt intra-protofilament subunit contacts, with subdomains 1 and 4. X-ray crystallography studies on the quadruple mutant (Supplementary Fig. 3a), using the cryoEM density map (Fig. 3a) as a molecular replacement model, led to the 3.25 Å structure of the monomer (G-ParM, Fig. 3b), which was subsequently used as a guide to construct the filament protomer structure in the cryoEM density (F-ParM, Fig. 3a–c and Supplementary Fig. 3b). Two of the mutation sites (Phe42 and Arg299), from the quadruple mutant, were observed to participate in intra-protofilament subunit contacts. This accounts for the success of the quadruple mutant in the preventing filament formation and allowing for monomer crystallization. The major difference between the monomer and filament protomers was the folding of a disordered region into a helix in F-ParM (F helix, Fig. 3b, c). This region is equivalent to the DNase I binding loop in actin, which is generally disordered in G-actin and becomes ordered to form a defined extended loop in F-actin (Fig. 3d, e and Supplementary Fig. 4)[15–17]. The nucleotide-binding site in the pCBH G-ParM and F-ParM structures contain ADP and the magnesium ion is coordinated by Asp9, Asp195, and Gln168 equivalent to Asp11, Asp154, and Gln137 in actin, suggesting a common mechanism of ATP hydrolysis in which Gln168 (Gln137) directs the hydrolytic water towards the ATP γ-phosphate for nucleophilic attack (Supplementary Fig. 4). Actin residue His161, which has also been proposed to be the catalytic

base in ATP hydrolysis[18], is not conserved in the pCBH ParM structure and has the opposite charge, Asp202 (Supplementary Fig. 4), indicating that the conserved glutamine may represent the evolutionary conserved residue important for hydrolysis[19]. Despite the similarities to actin, pCBH ParM is a genuine ParM, since it has a closed β-barrel, a feature that is missing from actin and MreB structures (Fig. 4). Thus, at the protomer level, pCBH ParM adopts a fold that is typical for ParMs, which themselves are a subset of the actin superfamily fold.

**pCBH ParM filament structure**. By contrast, the pCBH ParM filament structure is far more complex in comparison to the known filament structures of actin, MreB, and other ParMs (Fig. 5 and Supplementary Movie 1). The pCBH ParM filament has a diameter of 26 nm and of consists of 15 loosely associated, left-handed helical, single strands arranged in three layers in cross-section (Fig. 5a–c). At the center there is a single straight helical strand. This is surrounded by an intermediate layer of six interacting twisted helical strands that have the opposite polarity (antiparallel) to the central strand. The outer layer comprises eight non-interacting helical strands, which have a different twist and polarity (antiparallel) to the intermediate layer strands but have the same polarity (parallel) as the central strand (Figs. 5 and 6a–c). The 15 strands are in register in each cross-section despite having different radii and path lengths (Fig. 6d). This architecture

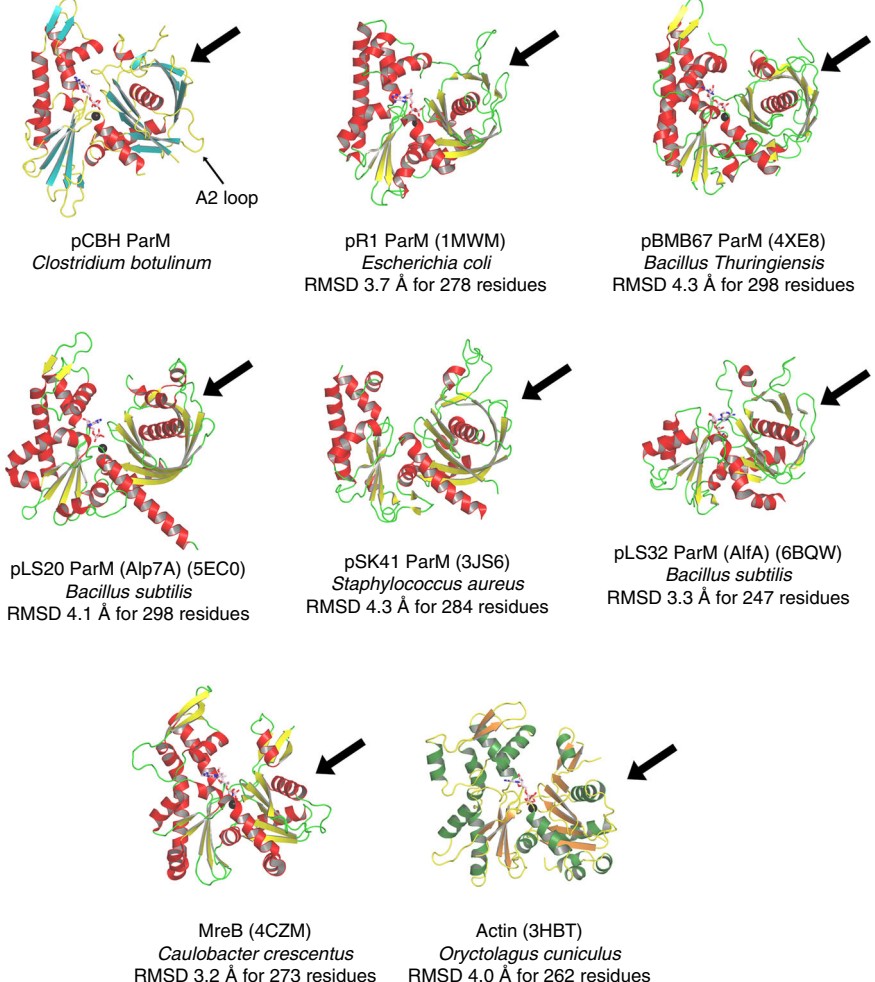

**Fig. 4** Comparison of the pCBH ParM structure with other filament-forming actin-like proteins. The thick arrows point towards the closed β-barrel, which is a feature of plasmid segregating ParMs and is absent in MreB and actin. This arrow also points towards the A1 loop in the ParMs. The extended A2 loop, a distinctive feature of pCBH ParM, is indicated

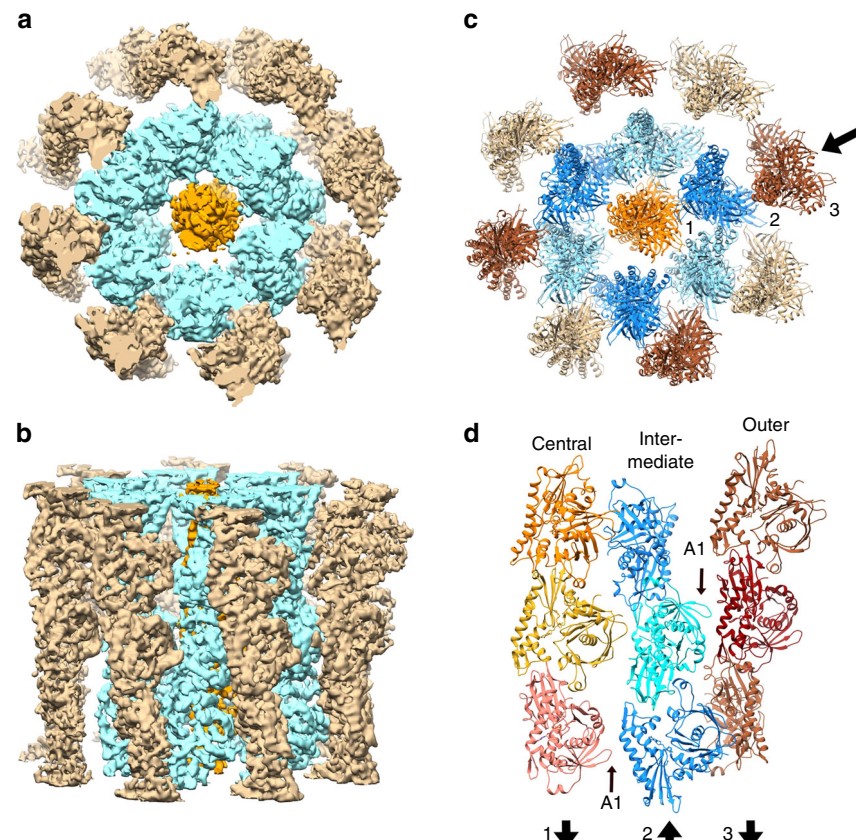

**Fig. 5** pCBH ParM filament structure. **a** End and **b** side views of the cryoEM density map at 4.7 Å resolution. **c** End view of fitted model showing the 15 strands. The layers in each lateral cross-section are defined as increasing in radius from the central strand (shown in orange), the intermediate layer (six strands in blue), to the outer layer (eight strands in brown). **d** Three strands indicated by the black arrow in **c**. The polarity of each layer is indicated by the lower arrows, with the arrowhead indicating the direction of the "pointed" end in reference to actin[10]. Two acidic loops, that form inter-strand contacts, are labeled (A1). The construction of the filament is detailed in Supplementary Movie 1

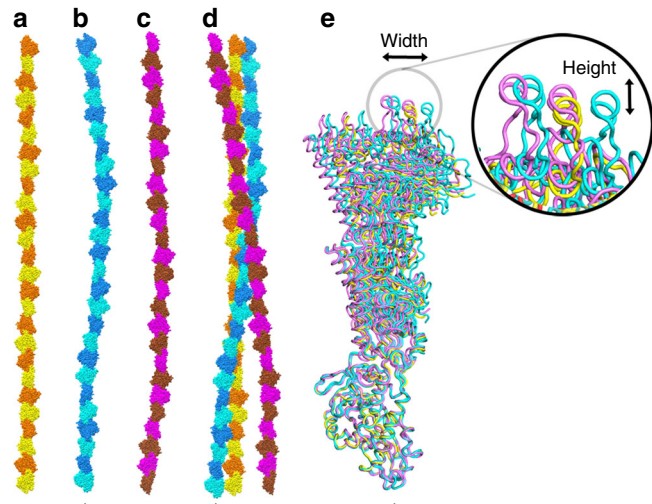

**Fig. 6** pCBH ParM protofilament construction. **a** The central strand. **b** An intermediate layer strand. **c** An outer layer strand. **d** Strands (A–C) as they are arranged in the filament. **e** Overlay of five protofilaments containing three protomers. The central strand (yellow), and two strands from the intermediate (blue) and outer (purple) layers that show the greatest divergence. The strands are aligned based on the lower protomers (barbed end with respect to actin)[10]. The arrows and enlarged section highlight differences in position in the *x*- and *y*-axes

is completely different to the *E. coli* R1 plasmid ParM filament structure, in which two protofilaments tightly associate in a staggered, parallel arrangement[2]. Furthermore, two R1 ParM filaments can come together in an antiparallel, staggered architecture[2]. None of the R1 ParM inter-protofilament interactions resemble those observed in the pCBH ParM filament.

**Intra-strand contacts.** Next, we examined the pCBH ParM filament structure to understand how a single protomer forms strands of different lengths and twists (Fig. 6e). The interactions between intra-strand protomers fall into two categories (Fig. 7a), and involve similar contact regions to actin intra-strand contacts[15] (Fig. 7b). Firstly, the central interaction region involves long-chain charged and polar residues between subdomains 2 and 4 from the lower protomer and subdomain 3 from the upper protomer (Fig. 7a and Supplementary Fig. 3a). Secondly, hydrophobic residues on the F helix from the subdomain 2 of the lower protomer fit into the hydrophobic pocket between subdomains 1 and 3 of the upper protomer (Fig. 7a). The hydrophobic residues in this interaction have fewer rotatable bonds than in the central interaction site (Fig. 7a and Supplementary Fig. 3a). This architecture allows the hydrophobic interaction region to act like a ball-and-socket while the central interaction region is elastic due to the flexible long-chain interactions. The linker residues on either side of the F-helix are relatively flexible (Supplementary Fig. 5a), which adds further pliancy into the intra-strand contacts. Taken together, these features result in a malleable protofilament that can be twisted and extended to adopt the different strand

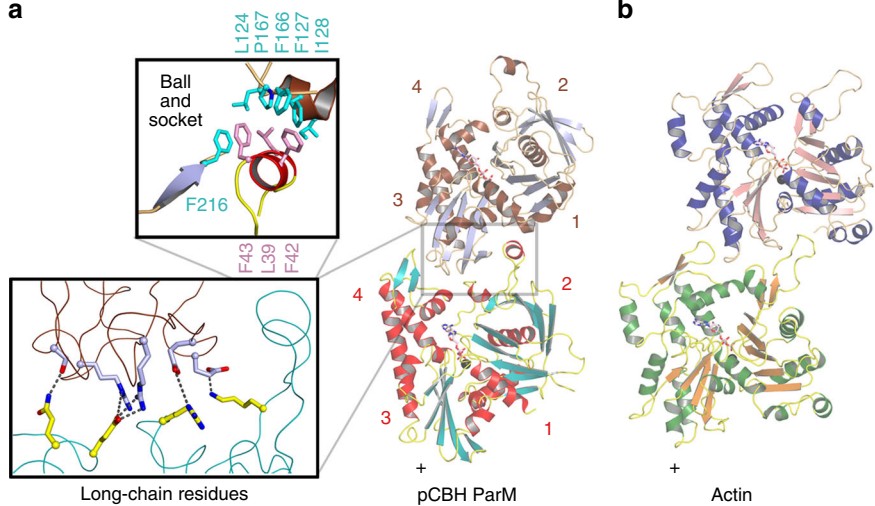

**Fig. 7** Interactions in the pCBH ParM protofilament. **a** Longitudinal interaction between two protomers in a strand. Blow-ups show the central electrostatic long chain interactions (lower) and the ball-and-socket hydrophobic interactions (upper). The subdomains are numbered. **b** Longitudinal interaction between two protomers in an actin strand[15]

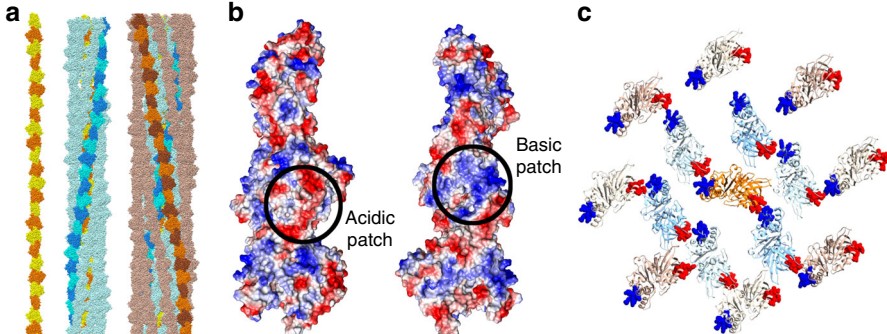

**Fig. 8** Charge–surface interactions between pCBH ParM filament layers. **a** The three layers. Central (yellow/orange), intermediate (blue/cyan), outer (brown/orange). **b** Charge surface of the protofilament comprised of three protomers. **c** Relative position of the basic and acidic patches in a single cross-section

conformations observed in the three-filament layers of strands (Fig. 6).

**Inter-strand contacts**. Finally, we inspected the pCBH ParM filament structure to determine the mechanism by which a single protomer forms many types of inter-strand interaction between strands of varying twist (Fig. 8a). The major type of inter-strand interaction occurs between complimentary basic and acidic patches on the protomers (Fig. 8b, c). Prominent in the acidic patch are two acidic loops (A1 loop and A2 loop, Figs. 3b, 5d and Supplementary Fig. 4), which adopt different positions across the basic surfaces of neighboring protomers in a lateral cross-section (Supplementary Figs. 5 and 6), allowing for multiple types of inter-strand contacts. The A1 loop is missing from the actin structure (Figs. 3, 4 and Supplementary Fig. 4) and involved in minor contacts between the two parallel strands of the ParM R1 filament[2]. The A2 loop is extended in pCBH ParM relative to actin and ParM R1 (Fig. 4). This loop lies on the outside of the actin filament[15] and forms some secondary contacts between ParM R1 antiparallel filament doublets[2]. Thus, these loops are not used in assembling the actin filament and play minor roles in assembling ParM R1 filaments and filament doublets.

The helicity of the pCBH ParM central strand dictates the interactions between the intermediate and outer strands through twisting ~50° per cross-section (compare the central strand with the blue and red protomers in Fig. 9 and Supplementary Fig. 7). In a single cross-section, the central strand forms significant antiparallel interactions with two opposing strands from the pseudo-hexagonal intermediate layer (Supplementary Figs. 6 and 7). The remaining four intermediate layer strands form two pairs of parallel interactions that have no or little interaction with the central strand. Six of the eight outer strands form antiparallel interactions with the intermediate layer strands (Supplementary Figs. 5 and 7). Two outer layer strands form no contacts with the intermediate layer strands. Directly across the pseudo-octagonal outer layer from these two strands are two outer layer strands that each form antiparallel interactions with two intermediate layer strands. Thus, on any one cross-section, some stands have strong inter-strand contacts, while others do not contact their neighbors (asterisks in Fig. 9a–c). However, the strands in each layer experience the same range of contacts over the 360° twist of the central protofilament (Supplementary Fig. 7 and Supplementary Movie 2). This design allows the 15 malleable strands to assemble into a stiff filament.

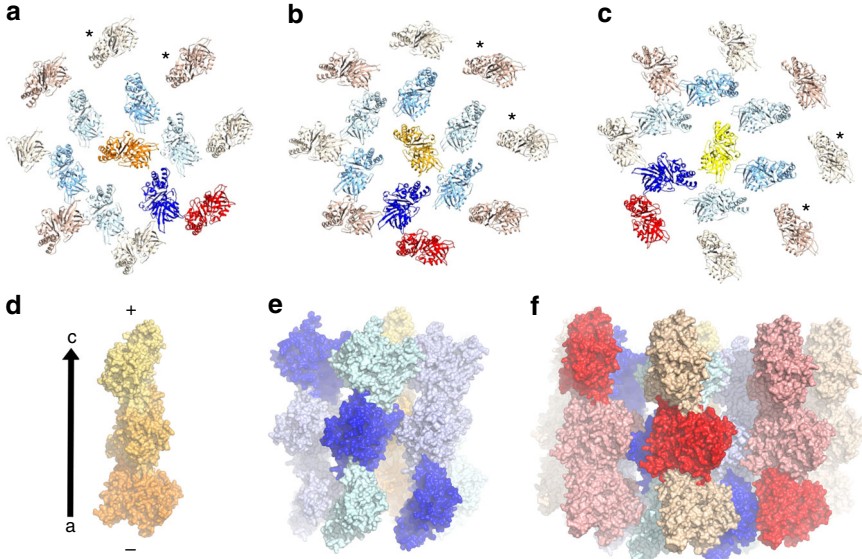

**Fig. 9** Arrangement of the pCBH ParM filament cross-sectional layers. **a–c** Three consecutive lateral cross-sections is moving towards the barbed end of the central strand. Subsequent cross-sections are found in Supplementary Fig. 7 and Supplementary Movie 2. Asterisks indicate protomers that have no inter-strand contact in the cross-section. The contribution of the central strand helicity to the filament architecture is highlighted by the blue and red protomers. **d–f** The blue and red protomers are contributed from neighboring strands in adjacent cross-sections

## Discussion

In summary, we have demonstrated an interaction between pCBH ParR and pCBH *parC* and have determined the structure of the pCBH ParM polymerizing system. The characteristics of the pCBH ParMRC elements allow for a hypothesis for the mechanism of pCBH plasmid segregation in *C. botulinum* (Fig. 10). We propose that multiple copies of the adaptor protein ParR bind to the *parC* repeats on the pCBH plasmid in a circular arrangement, as determined for the pSK41 and R1 plasmids[20,21]. ParR from the R1 plasmid has been shown to bind to the barbed end of ParM monomers and associate with barbed ends of ParM filaments[22]. The pCBH ParM filament presents different numbers of barbed end protomers at the two ends of the filament, 6 and 9, which include two circular arrangements of 6 and 8 barbed end protomers. Thus, two copies of the pCBH plasmid can be expected to bind to a single elongating bipolar 15-stranded pCBH ParM filament, one at each end, via some of the copies of the circularly arranged plasmid-bound ParR. The remaining copies of plasmid-bound ParR bind to ParM monomers. Elongation of the pCBH ParM filament ends at the attachment site of the two plasmids will then proceed via the ParR-bound ParM monomers joining the filament with the release of the filament-bound ParRs. In this mechanism, the plasmids remain attached to the filament and ParM polymerization provides the force to drive the plasmids to the two extremes of the *C. botulinum* cell. The pointed ends of strands would elongate by incorporating free pCBH ParM to match the barbed-end strands' elongation rates, to maintain the integrity of the filament.

The pCBH ParM filament appears to be highly adapted to segregating two large plasmids, which are 245–267 kb in size for plasmids containing this ParMRC system. In comparison, the *E. coli* R1 plasmid is 98 kb in size and is segregated by the anti-parallel association of two 2-stranded ParM filaments[2,23]. The pCBH ParM "actin-like" filament has a similar number of pro-tofilaments and diameter (26 nm) to chromosome-segregating microtubules[24]. The pCBH ParM filament is bipolar and stiff, and like microtubules, the 15 polymerizing strands are likely to exert greater combined force relative to typical two-stranded actin-like filaments.

## Methods

**Protein expression and purification**. Constructs of ParM (AJD29063.1) and ParR (AJD29378.1) from plasmid pCBH (CP006901.1) were synthesized and cloned into pSY5, a modified pET-21d(+) (Novagen) plasmid encoding an 8-histidine tag, followed by a HRV 3C protease cleavage site, ahead of the N-terminus of the proteins[25]. Plasmids were transformed into BL21 (DE3) cells, grown to OD$_{600}$ of ~0.8 and protein expression was induced with 1.0 mM IPTG overnight at 15 °C. The cultures were then centrifuged at 4000×*g* and the cell pellets resuspended in 50 mM Tris–HCl, pH 8.0, 500 mM NaCl, 20 mM imidazole, 5% glycerol, 0.5 mg/ml lysozyme and protease inhibitor tablets (1 per 2 l culture, Roche) and lysed via sonication. The cell lysate was then clarified via centrifugation at 30,000×*g* for 30 min and loaded onto a HisTrap FF 5 ml (GE Healthcare) pre-equilibrated with 50 mM Tris–HCl, pH 8.0, 500 mM NaCl, and 20 mM imidazole. Following loading and washing of the column, HRV 3C protease was loaded in the same buffer for cleavage of tagged proteins over 12 h at 4 °C. The cleaved protein was then eluted, pooled, and subjected to size-exclusion chromatography (Superdex 75 pg, GE Healthcare) in 40 mM HEPES, pH 7.5, 150 mM KCl, 2 mM MgCl$_2$, and 1 mM DTT. Fractions were checked for purity via SDS–PAGE, and the pure fractions pooled and concentrated to between 5 and 10 mg/ml as determined by UV absorbance at 280 nm using an estimated A280 value calculated using PROTEIN CALCULATOR v3.4 (http://protcalc.sourceforge.net).

**Polymerization and phosphate release assays**. Assembly of pCBH ParM and the 1:0.5 ParM/ParR complex was initiated by addition of 2 mM ATP at 24 °C in 20 mM HEPES, pH 7.5, 350 mM KCl, 2 mM MgCl$_2$, and followed by light scattering (600 nm) at 90° using either a Perkin Elmer LS 55 spectrometer. The release of inorganic phosphate (Pi) upon nucleotide hydrolysis during pCBH ParM poly-merization was measured at 24 °C using the Phosphate Assay Kit (E-6646) from Molecular Probes. The absorbance at 360 nm was measured using Ultraspec 2100 pro (Amersham Biosciences).

**Electrophoretic mobility shift assays**. The standard reaction mixture (10 μl) for the pCBH *parC*/ParR experiments contained 2–320 μM of ParR, 40 mM HEPES, pH 7.5, 300 mM KCl, 2 mM MgCl$_2$, 1 mM DTT, 1 mg/ml bovine serum albumin, 0.1 μg/μl sonicated salmon sperm DNA and 5% glycerol. The 5′ 6-FAM-labeled DNA fragment was generated via PCR from a pCBH *parC* (Supplementary Fig. 3c) encoding pUC57 plasmid synthesized by GenScript (Singapore). The primers were 5′ 6-FAM-M13-F and M13-R (Supplementary Table 3). 5′ 6-FAM is a single isomer derivative of fluorescein. The standard reaction mixture was mixed at room temperature for 10 min, followed by the addition of 5′ 6-FAM-labeled DNA fragment and further incubation of 20 min. 20 nM of 5′ 6-FAM labeled DNA fragment was used in all experiments. Polyacrylamide gels were pre-run at 150 V for 1 h. After incubation, reactions were analyzed by electrophoresis on a 1 × TBE, pH 7.5, 4% polyacrylamide gel in 1 × TBE running buffer (0.89 M Tris–base, 0.89 M boric acid, 0.02 M EDTA, pH 8.3) at 150 V for 1 h. Gels were then scanned using a Pharos FX Plus Molecular Imager (Bio-Rad), which enabled visualization of the 5′ 6-FAM-labeled *parC* DNA.

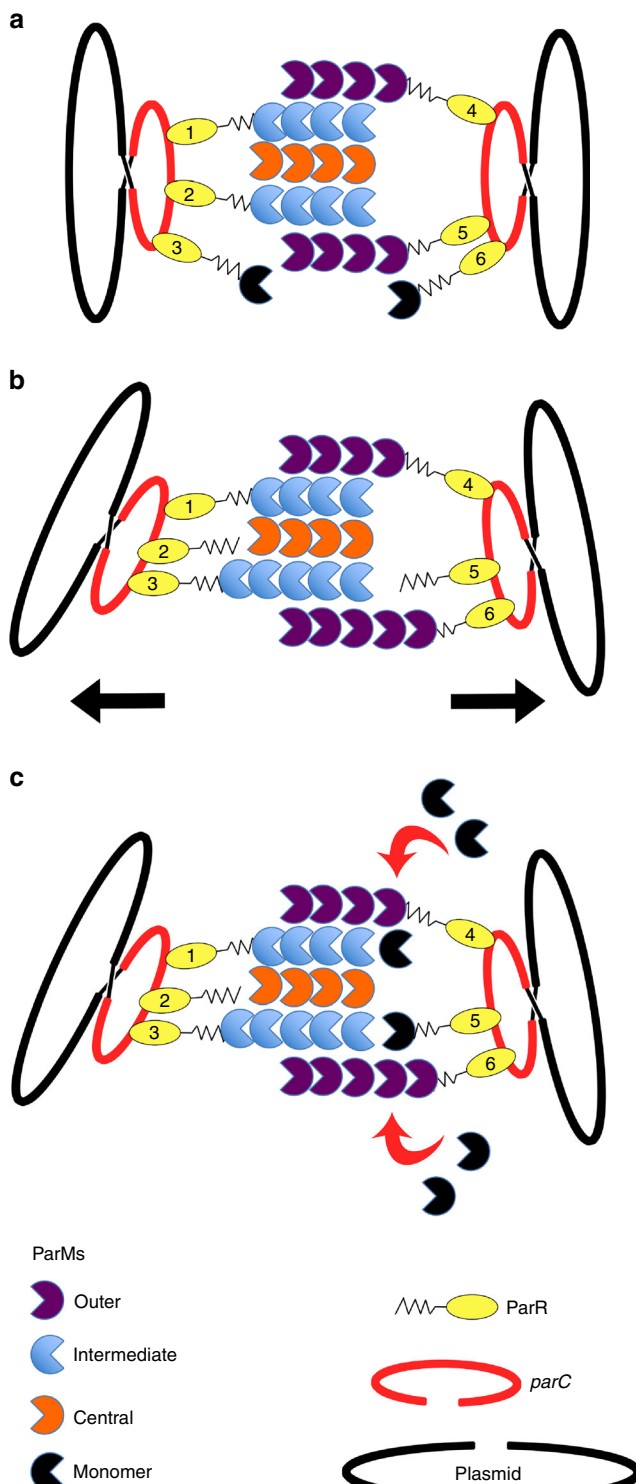

**ParMs**

Outer

Intermediate

Central

Monomer

ParR

*parC*

Plasmid

**Fig. 10** Hypothetical model for pCBH plasmid segregation. **a** A copy of the plasmid is attached to both ends of the filament via ParRs (labeled 1, 2, 4, and 5), linking the "barbed" ends of ParM strands to *parC* (red). ParR-bound (labeled 3 and 6) ParM monomers are shown in black. **b** Release of ParRs (labeled 2 and 5) from the filament allows for new ParR-bound (labeled 3 and 6) ParM subunits to be added to the filament, while maintaining filament/plasmid contacts through other ParRs (labeled 1 and 4). **c** "Pointed" end addition to ParM strands and rebinding of ParR (labeled 5) to ParM is provided from the pool of unassociated ParM monomers. For clarity, this figure illustrates a subset of the ParM filament strands and *parC*-bound ParRs

**Sample preparation for EM**. Frozen ParM protein was quickly thawed then diluted into polymerization buffer (40 mM HEPES, pH 7.4, 70 mM KCl, 7 mM MgCl₂, 1 mM DTT) to 20 μM. ATP was then added to a final concentration of 3 mM and the reaction mixture allowed to incubate for 30 min at 25 °C, to allow the filaments to form.

**Transmission electron microscopy of negatively stained specimens**. STEM100Cu Elastic carbon substrate grids (Ohkenshoji Co., Ltd.) were glow discharged (5 mA, 45 s) and used within one hour. 5 μl of the incubated reaction mixture was applied to the glow discharged grids and allowed to absorb for 1 min. The reaction mixture was then blotted away using filter paper and 5 μl of 1% uranyl acetate applied to the grid. Staining was allowed to proceed for 1 min before being blotted by filter paper. Prepared grids were then dried overnight in a dry box before imaging. Grids were observed with a Hitachi H-7600 electron microscope operated at 100 kV and at a nominal magnification of ×40,000. Films were scanned in 7 μm steps with a Zeiss Z/I Imaging PhotoScan 2000 scanner. Fifty filament images were extracted, unbent and averaged. A Fourier pattern was calculated. Many obvious layer lines were observed, indicating the filament structure is rigid and uniform.

**Cryo-electron tomography and helical parameters determination**. BSA-coated colloidal gold in the polymerization buffer (40 mM HEPES, pH 7.4, 70 mM KCl, 7 mM MgCl₂, 1 mM DTT, and 20 μM ATP) was prepared by mixing 5 nm colloidal gold (Sigma) and BSA (Sigma)[26]. MultiA Cu 200 grids (Quantifoil) were glow discharged and used within an hour after discharging. 1.5 μl of the polymerized ParM solution and 1.5 μl of the colloidal gold solution were applied on the glow discharged grid, blotted for 5 s on the EM GP (Leica) and plunge vitrified using liquid ethane cooled by liquid nitrogen. Frozen grids were kept under liquid nitrogen for no more than 1 week before imaging.

Data series for cryo-electron tomography was acquired with a Tecnai G2 Polara (FEI, Nagoya University) equipped with a FEG operated at 300 kV, CCD camera (Gatan US 4000) and an energy filter (Gatan GIF). Tilt series were recorded with 2° increment from −60° to 60°, with the energy filter using a slit width of 20 eV, with −5 μm defocus and 1 e⁻ dose per Å² per one image. Tomogram reconstruction was performed using IMOD[27]. Five tomograms were imaged and assembled. The tomogram with the most distinct layer line pattern (Supplementary Fig. 2) was helically averaged in real space using different helical parameters. One helical parameter was selected (−50.1° twist/52 Å rise) which showed the best match with the averaged layer lines (Fig. 2b).

**Cryo-transmission electron microscopy**. R1.2/1.3 Mo400 grids (Quantifoil) were glow discharged and used within an hour. 2.5 μl of the incubated reaction mixture was applied on the glow discharged grids, blotted for 5 s on the EM GP (Leica) and plunge vitrified using liquid ethane cooled by liquid nitrogen. Frozen grids were kept under liquid nitrogen for not more than 1 week before imaging. For screening of good conditions for cryoEM imaging, grids were manually observed in a Tecnai G2 Polara (FEI, Nagoya University) equipped with a FEG operated at 300 kV and a minimal dose system. Images were captured at a nominal magnification of ×115,000 with an underfocus ranging from 1.5 to 3.5 μm and by subjecting the sample to a 2 s exposure time totaling ~30 electrons per Å² of electron dose. Images were recorded on a GATAN US4000 CCD camera which was energy filtered by an in-column energy operated between 10 and 15 eV filter width with each pixel representing 1.8 Å at the specimen level at exposure settings. Grids for data collection were made via the Spotiton system[28,29]. For data collection, grids were semi-automatically imaged using Appion on a Titan Krios (New York Structural Biology Center) equipped with a FEG operated at 300 kV and a minimal dose system. Images were captured at a nominal magnification of ×91,000 with an underfocus ranging from −1 to −2.5 μm for 14 s of exposure time totaling ~54 electrons per Å² of electron dose over 70 frames on a K2 Summit with each pixel representing a calibrated 1.331 Å at the specimen level.

**Image analysis**. 1786 raw 'movie' cryoEM data were collected and processed in RELION 2.1[13,14]. Drift correction was carried out using MotionCor2[30], and the CTF for each micrograph calculated via CTFFIND4.1[31,32]. Filaments were manually picked with e2helixboxer from the EMAN2 software package[33]. 36,292 particles were extracted at a box size of 320 × 320 pixels. After 2D classification, 33,356 particles were selected. The helically averaged cryo-electron tomogram was used as the initial reference. Helical symmetry converged to −50.4° twist and 50.3 Å rise and the resolution reached 4.7 Å. Frames 3–42 of the raw movies were used for the final reconstruction, and the total dose was ~30 electrons per Å².

**Strand averaging in real space**. Local resolution was calculated by ResMap[34] (Supplementary Fig. 2b) and the intermediate layer showed the best resolution. To achieve higher resolution, the six strands in the intermediate layer were averaged in the real space. The region between 2.4 and 8.8 nm of the radius from the filament axis was extracted from half maps as the intermediate layer. The intermediate layer was segmented into six strands. The density for these six strands were aligned and averaged. The resolution of the averaged map (4.2 Å) was evaluated as a mean resolution by ResMap (Supplementary Fig. 2c).

**Persistence lengths of the pCBH ParM and actin filaments**. Eighteen and 21 cryoEM filament images, which were longer than 350 nm, were randomly selected from the pCBH ParM data (Fig. 2c) and from published actin filament[35] cryoEM images, respectively. Filament images were manually tracked with NeuronJ (https://imagescience.org/meijering/software/neuronj/) and analyzed as outlined by Ott et al[36]. $\vec{t}(s)$ is the unit vector representing direction of the filament at position $s$, the coordinate along the filament track. The average of inner product,

$$\vec{t}(s) \cdot \vec{t}(s+x) \qquad (1)$$

is plotted against distance $x$ (Supplementary Fig. 8). The persistent length ($L_p$) was calculated by fitting

$$\langle \vec{t}(s) \cdot \vec{t}(s+x) \rangle = \exp\left(-\frac{x}{2L_p}\right) \qquad (2)$$

**Crystallization and monomer X-ray structure determination**. The DNA encoding a quadruple mutant in the polymerization interfaces of pCBH ParM (Supplementary Fig. 3a) was generated via two rounds of PCR using primers: F42D_147D_F and F42D_147D_R; and S298D_R300D_F and S298D_R300D_R (Supplementary Table 3). The resultant purified protein subjected to crystallization trials. 5 mM ATP was added to 5 mg/ml pCBH ParM mutant and incubated for 1 h on ice. Crystals were grown via hanging-drop vapor diffusion in drops containing 0.5 μl of protein/ATP and 1 μl of mother liquor (28% PEG 400, 0.1 M HEPES, pH 7.5, 150 mM MgCl₂) at 288 K. Native X-ray diffraction data were collected on a RAYONIX MX-300 HS CCD detector on beamline TPS 05A (NSRRC, Taiwan, ROC) controlled by BLU-ICE (version 5.1) at $\lambda = 1.0$ Å. Data were indexed, scaled, and merged in HKL2000 (version 715)[37]. Molecular replacement using the pro-tomer cryoEM density map was carried out in the Phaser[38] followed by building in Coot[39] and refinement in PHENIX[38]. The model was built into the three-fold averaged electron density map. Group B factors and three-fold non-crystallographic symmetry constraints were maintained until the final round of refinement. The model quality was checked with the MolProbity[40]. Data collection and final refinement statistics are summarized in Supplementary Table 1.

**Model building for a single strand**. The missing loop (38–45) in the crystal structure of pCBH ParM was modeled by Rosetta3. Two subunits of the resultant model were fitted into the averaged strand density map and then refinement into the density map was performed by Rosseta3 and then PHENIX[41]. The two subunits were subjected to symmetry restraints in the refinement.

**Model building for the entire structure**. Fifty-one subunits of the model for the averaged strand were fitted to three adjacent slices of the whole 4.7 Å map. Each subunit was fitted individually. The atomic models were flexibly fitted to the density map with MDFF on NAMD 2.12 software[42]. The calculation was performed under harmonic helical symmetry restraints on all atoms except hydrogens, and the solvent effect was modeled with generalized Born implicit solvent. In the MDFF, the main chain near the interface with the other subunits was allowed to move while other regions were fixed.

**Reporting summary**. Further information on research design is available in the Nature Research Reporting Summary linked to this article.

## Data availability

The atomic coordinates and structure factors have been deposited in the Protein Data Bank under the accession code 6IXW. The coordinates and cryoEM maps for pCBH F-ParM (accession codes 6IZR and EMD-9757) and the averaged strand from the intermediate layer (accession codes 6IZV and EMD-9758) have been submitted to the Protein Data Bank and Electron Microscopy Data Bank, respectively. Other data are available from the corresponding authors upon reasonable request.

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

## Acknowledgements

We acknowledge A*STAR, RIIS, JSPS KAKENHI (18H02410 to A.N.) for funding. We thank Professor Koh Saito and Associate Professor Makoto Kuwahara of Nagoya University, Japan for use of the FEI Polara for screening cryoEM conditions. We thank the Synchrotron Radiation Protein Crystallography Facility of the National Core Facility Program for Biotechnology, Ministry of Science and Technology and the National Synchrotron Radiation Research Center, a national user facility supported by the Ministry of Science and Technology, Taiwan, ROC; We thank Bridget Carragher and Clint Potter for use of the Simons Electron Microscopy Center and National Resource for Automated Molecular Microscopy located at the New York Structural Biology Center, supported by grants from the Simons Foundation (SF349247), NYSTAR, and the NIH National Institute of General Medical Sciences (GM103310) with additional support from Agouron Institute (F00316) and NIH (OD019994).

## Author contributions

F.K., A.N., D.P. and R.C.R. conceived experiments, performed experiments, analyzed data, and wrote the paper. L.J.L. prepared reagents. K.T. analyzed data. V.P.D. froze cryoEM grids. Y.Z.T. helped in cryoEM data collection.

## Additional information

**Competing interests:** The authors declare no competing interests.

