## [Peer Review File · Nature Communications]

Reviewers' comments:

Reviewer #1 (Remarks to the Author):

The huge polymer composed of 15 filaments of an actin-family protein is striking, a novel result well worth publishing. However, the paper does not explain the conditions where these 15-strand polymers form. Neither the main text nor the supplementary figures reveals the homogeneity of the samples. Are these 15-strand polymers assembled in living cells and used to segregate plasmids? (They should be much easier to image than more conventional actin filaments.)

The short format of the manuscript makes the reporting of the results very terse. To note two examples among many, "Addition of pCBH ParR to the pCBH ParM resulted in faster assembly and higher levels of light scattering." How much faster? Does this depend on the concentration of ParR in any way that is informative about the mechanism? The phase "defined mobility shift" needs the addition of "native gel electrophoresis," "with DNA fragments were generated via PCR from pCBH parC" and the direction of the shift. What is "FAM-labelled DNA fragments?" What was imaged when the gels were scanned?

Fig. 1: A, calculating the rate constant for phosphate release would be easy and good to know. Please specify the nucleotide and divalent cation in each structure including actin. What is the difference between the resolution of 4.2 Å on page 3 and the resolution of 4.7 Å given in the methods section?" Is there a histidine equivalent to H161, the other proposed catalytic residue in conventional actin filaments?

"Negative stain electron microscopy" is a misnomer; it is "transmission EM of negatively stained specimens."

Page 4: I am skeptical that "plastic" is the right word in the following: "central interaction region is plastic due to the flexible long-chain interactions". Do you mean elastic?

The section heading "Extended Data Table 1. X-ray data collection and refinement statistics" should be on page 10 of the extended data document. Explain while some of the parameters are in parentheses.

I expected to see a table with a summary of the EM statistics comparable to Table 1.

Tom Pollard

Reviewer #2 (Remarks to the Author):

This study by Koh et al. is quite interesting and experimentally well executed, and I would like to recommend publication in NC. It describes the unexpected finding that the bacterial actin-like protein ParM from *Clostridium botulinum* forms a filament consisting of 15-strands with mixed polarity, distinct from the polar double-helix formed by eukaryotic actin. ParM is one of three components of ParMTC, a plasmid segregation motor, which additionally comprises a centrosome-like DNA sequence (parC) and an adaptor protein (ParR) that is responsible for linking the plasmid to the growing ends of the ParM filament for segregation. The ParMRC cassette from *Clostridium botulinum* drives the segregation of the large, 257 kb pCBH plasmid, which carries the botulinum neurotoxin type B.

Koh et al. describe the cryo-EM structure of the ParM filament at 4.7 Å resolution and the crystal structure of a non-polymerizable ParM mutant. While these two structures constitute the bulk of the work, they also present light scattering data that demonstrates the ability of this system to act as a segregation motor stimulated by pCBH.

My sole major concern is with the way in which the data is discussed and illustrated. It seems that the paper was originally prepared for a smaller format journal. Both the text and figures can be

significantly improved, and in most cases expanded to improve clarity for NC. I am aware that this is too general a comment, and I try to provide some specific examples below to help the authors understand this point. I have read several papers from these authors, and they read very well, so I am sure they will be able to make the necessary changes.

Specific examples:

a. The biochemical data in Extended Data Figure 1 (which should probably be renamed as Supplementary Figure 1) should be probably presented in the main paper, since it makes a very important functional point about the ability of ParMRC to function as a segregation motor, and would also improve the balance between structure and function in the paper. The significance of the findings for plasmid segregation should be also more extensively discussed. For instance, what is the meaning of a critical concentration of 3 μM ? A priori, this value seems high compared to eukaryotic actin. Is this sufficient to drive plasmid segregation in *Clostridium botulinum*? What is the concentration of ParM in the cell? These are a few examples of how the results could be better conceptualized, which would improve the paper immensely.

b. Figures 2 to 4 of the main paper would all benefit from larger panels and more labels. I actually could not understand the nature of the various inter-subunit contacts that hold this 15-stranded filament together. And these contacts should be properly classified as longitudinal, lateral, layer 1, 2 and 3. For instance most of Fig 4c is too small to understand, and there are only two labels. Video S2 runs too fast, and also would benefit from labels.

c. It would also help if a figure were to be added with a functional model of the ParMRC segregation system based on the structural data presented here and previously published.

d. p3 "increasing levels of pCBH ParR resulted in a defined mobility shift" Please be more specific about the concentrations of pCBH ParR used (both in the text and Extended Data Figure 1).

e. What was the rationale for the design of the non-polymerizing quadruple mutant of ParM (based on what is known about the contacts in the filament?)

f. The abstract and text state that the ParM filament is stiffer than the actin filament. What is the evidence for this? Persistence length measurements? Or is this simply based on visual inspection by cryo-EM? This is an important point, but needs experimental support.

g. p3 "the metal ion" What is the ion?

These are just but a few examples of how the text and Figures could be improved. Overall, however, this is an important contribution suitable for publication in NC

Roberto Dominguez

Reviewer #3 (Remarks to the Author):

The manuscript by Koh et al provides an extensive study of the structure of the 15-strand parM filament from *Clostridium botulinum*. Despite a large number of experiments that yielded high resolution structure of the polymer the paper lacks crucial information on methods and key parameters of the pCBH parM filament. The lack of any useful discussion contributes to my pessimism regarding publishing this paper in its current form in any high profile journal. I urge the authors to constructively resolve all the issues appended below:

1. The intensity of light scattering upon pCBH polymerization in Fig 1A does not match the intensities shown in SI Fig 1B. For example, the intensity obtained for 20 μM ParM in SI Fig 1B is lower than that for the 15 μM parM shown in Fig 1A. Therefore, it is not clear which intensities were used in the plot in Fig 1B to calculate critical concentration.

2. Scale bars in SI Fig 1F are unreadable!

3. There is a very poor match between the layer lines obtained from the negatively stained and frozen parM filaments in SI Fig 2. In line with that, the indexing of the layer lines is missing and

thus it is not clear how the helical parameters were determined. The layer lines should be indexed and corresponding helical lattice plot should be provided.

4. I'm skeptical that 4.2 Å resolution can be obtained from the tomogram series collected using regular CCD camera. It is even more puzzling that from the data collected using direct detector on Titan Krios TEM the authors obtained 4.7 Å map. On top of that, it is not clear whether all frames were used for 3D-reconstruction (with a dose of 54 e/ Å²!) or dose fractioning was used.

5. Regarding the reconstruction approach - the methods section does not provide any information on whether the point group symmetry was used. I assume that the helical parameters provided for "slices" correspond to a point group symmetry rotation and translation. It is also unclear how the averaged strand (SI Fig 3) was obtained taking in account that individual protofilaments forming the central, intermediate and outer layers have different helical arrangements which are not spelled out in the manuscript. It is also not clear how these symmetries are related to the helical parameters found for the parM filament earlier (Galkin et al., 2009; Bharat et al., 2015). The interface between parM protomers described in the paper should be carefully compared with the previously published data (Bharat et al., 2015).

6. The authors do not discuss how the inter-strand contacts found in the pCBH filament are related to the interactions observed between parM doublets formed in presence of PEG 6000 (Bharat et al., 2015).

7. The manuscript missing crucial discussion regarding how the observed structure of the pCBH filament may promote segregation of extra-large plasmids and what is the difference in the size of plasmids between different bacteria that use parM-like proteins for plasmid segregation.

Reviewer #1 (Remarks to the Author):

The huge polymer composed of 15 filaments of an actin-family protein is striking, a novel result well worth publishing. However, the paper does not explain the conditions where these 15-strand polymers form.

We thank Professor Pollard for his comments in order to improve the manuscript. We apologise of the oversight in not reporting the filament conditions. We now include: “The pCBH ParM filaments could be imaged under a wide range of conditions including high physiological salt concentrations typically found in bacterial cells. The condition used to form the most homogeneous population for cryoEM imaging was 70 mM KCl, 7 mM MgCl₂, 2 mM ATP, 10 mM HEPES, pH 7.5.

Neither the main text nor the supplementary figures reveals the homogeneity of the samples.

We now state in the main text: “All filaments showed similar widths on the micrographs. We extracted 36292 particles and selected 33356 particles using Class2D in Relion^{14,15}, indicating more than 90% of the particles are homogeneous.”

Are these 15-strand polymers assembled in living cells and used to segregate plasmids? (They should be much easier to image than more conventional actin filaments.)

Imaging of the filaments in living cells is an exciting idea, unfortunately we don't currently have the facilities to grow the organism.

The short format of the manuscript makes the reporting of the results very terse. To note two examples among many, “Addition of pCBH ParR to the pCBH ParM resulted in faster assembly and higher levels of light scattering.” How much faster? Does this depend on the concentration of ParR in any way that is informative about the mechanism?

We agree that these are important questions that may give insight in to the mechanism. We originally did not undertake a titration of ParR in this experiment. Unfortunately, the laboratory has closed in the meantime, so currently we are not able to carry out the titration. Given that Reviewer 3 also had questions concerning this supplementary figure panel, we have removed it to avoid including incomplete data.

The phase “defined mobility shift” needs the addition of “native gel electrophoresis,” “with DNA fragments were generated via PCR from pCBH parC” and the direction of the shift.

Corrected as: “Titration of DNA fragments generated via PCR from pCBH parC with increasing levels of pCBH ParR resulted in a defined mobility shift to larger molecular size, consistent with a specific interaction between pCBH ParR and pCBH parC (Fig. 1d).”

What is “FAM-labelled DNA fragments?” What was imaged when the gels were scanned?

Thank you spotting the typo. We now include: “5' 6-FAM is a single isomer derivative of fluorescein.” and “Gels were then scanned using a Pharos FX Plus

Molecular Imager (Bio-Rad), which enabled visualisation of the 5' 6-FAM-labelled *parC* DNA.”

Fig. 1: A, calculating the rate constant for phosphate release would be easy and good to know.

We now include: “The Pi release rate was estimated from the linear slope to be ~ 10 nM/s” in the legend.

Please specify the nucleotide and divalent cation in each structure including actin.
Now included in the legend.

What is the difference between the resolution of 4.2 Å on page 3 and the resolution of 4.7 Å given in the methods section?

We failed to describe this properly. This has now been rectified: “These parameters refined to a distance 5.03 nm and twist -50.4 ° with the cryoEM data. Helical averaging of the cryoEM density, from each cross-section of the filament, based on these parameters led to a 4.7 Å map for the entire filament (Supplementary Figure 2a,b). Within each cross-section, an intermediate layer consisting of 6 hexagonal protomers showed the best local resolution. Inter-strand averaging for this intermediate layer led to a 4.2 Å map for the protofilament (Fig. 3a and Supplementary Figure 2).”

Is there a histidine equivalent to H161, the other proposed catalytic residue in conventional actin filaments?

Now we comment on this residue: “Actin residue His161, which has also been proposed to be the catalytic base in ATP hydrolysis¹⁹, is not conserved in the pCBH ParM structure and has the opposite charge, Asp202 (Supplementary Figure 4), indicating that the conserved glutamine may represent the evolutionary conserved residue important for hydrolysis²⁰.”

“Negative stain electron microscopy” is a misnomer; it is “transmission EM of negatively stained specimens.”

The error has been rectified.

Page 3: “Electron microscopy (EM) of negatively stained specimens”

Fig. 1: “EM image of a negative stained pCBH ParM filament”

Methods: “Transmission electron microscopy of negatively stained specimens”

Fig. 2b: “Averaged layer lines from 50 negatively stained filament images.”

Page 4: I am skeptical that “plastic” is the right word in the following: “central interaction region is plastic due to the flexible long-chain interactions”. Do you mean elastic?

Elastic is more appropriate. We have changed.

The section heading “Extended Data Table 1. X-ray data collection and refinement statistics” should be on page 10 of the extended data document. Explain while some of the parameters are in parentheses.

The section header is corrected and we include the sentence “Values in parenthesis refer to the parameters for the highest resolution shell.”

I expected to see a table with a summary of the EM statistics comparable to Table 1.

We now include the table. "Extended Data Table 2. CryoEM data collection, refinement and validation statistics."

Reviewer #2 (Remarks to the Author):

This study by Koh et al. is quite interesting and experimentally well executed, and I would like to recommend publication in NC. It describes the unexpected finding that the bacterial actin-like protein ParM from Clostridium botulinum forms a filament consisting of 15-strands with mixed polarity, distinct from the polar double-helix formed by eukaryotic actin. ParM is one of three components of ParMTC, a plasmid segregation motor, which additionally comprises a centrosome-like DNA sequence (parC) and an adaptor protein (ParR) that is responsible for linking the plasmid to the growing ends of the ParM filament for segregation. The ParMRC cassette from Clostridium botulinum drives the segregation of the large, 257 kb pCBH plasmid, which carries the botulinum neurotoxin type B.

Koh et al. describe the cryo-EM structure of the ParM filament at 4.7 Å resolution and the crystal structure of a non-polymerizable ParM mutant. While these two structures constitute the bulk of the work, they also present light scattering data that demonstrates the ability of this system to act as a segregation motor stimulated by pCBH.

We would like to thank Professor Dominguez for his positive comments.

My sole major concern is with the way in which the data is discussed and illustrated. It seems that the paper was originally prepared for a smaller format journal. Both the text and figures can be significantly improved, and in most cases expanded to improve clarity for NC. I am aware that this is too general a comment, and I try to provide some specific examples below to help the authors understand this point. I have read several papers from these authors, and they read very well, so I am sure they will be able to make the necessary changes.

We have expanded the figures in line with this comment. We have moved two figures from the extended data into the main manuscript, and split some of the main figures in to two.

Specific examples:

a. The biochemical data in Extended Data Figure 1 (which should probably be renamed as Supplementary Figure 1) should be probably presented in the main paper, since it makes a very important functional point about the ability of ParMRC to function as a segregation motor, and would also improve the balance between structure and function in the paper.

We have made these changes as suggested.

The significance of the findings for plasmid segregation should be also more extensively discussed. For instance, what is the meaning of a critical concentration of 3 μM? A priori, this value seems high compared to eukaryotic actin. Is this

sufficient to drive plasmid segregation in Clostridium botulinum? What is the concentration of ParM in the cell? These are a few examples of how the results could be better conceptualized, which would improve the paper immensely.

While we do not know the cellular concentrations of pCBH ParM, we address this question by comparison to E.coli R1 ParM. “The critical concentration for assembly was estimated to be around 3 μM from the plot of maximum intensity values of light scattering curves at different pCBH ParM concentrations (Fig. 1c). This compares with a similar value of 1.5-2 μM determined *in vitro* for the *Escherichia coli* R1 plasmid ParM¹⁰, for which the cellular concentration of ParM has been estimated to be 12-14 μM ¹¹. Thus, the filament assembly parameters are in line with this well characterized segregation system.”

b. Figures 2 to 4 of the main paper would all benefit from larger panels and more labels. I actually could not understand the nature of the various inter-subunit contacts that hold this 15-stranded filament together. And these contacts should be properly classified as longitudinal, lateral, layer 1, 2 and 3. For instance most of Fig 4c is too small to understand, and there are only two labels. Video S2 runs too fast, and also would benefit from labels.

We have added labels to Fig. 2 (now Fig. 5) and have split Figs. 3 (now Figs 6,7) and 4 (now Figs 8,9). We have defined the layers more precisely: “The layers in each lateral cross-section are defined as increasing in radius from the central strand (shown in orange), the intermediate layer (6 strands in blue), to the outer layer (8 strands in brown).” and have been more strict in using the term cross-section, and included longitudinal in the Sup. Fig. 7 legend. We have slowed down Video S7. We also include more description concerning the inter subunit contacts: “In a single cross-section, the central strand forms significant antiparallel interactions with two opposing strands from the pseudo-hexagonal intermediate layer (Supplementary Figure 6-7). The remaining four intermediate layer strands form two pairs of parallel interactions that have no or little interaction with the central strand. Six of the eight outer strands form antiparallel interactions with the intermediate layer strands (Supplementary Figure 5,7). Two outer layer strands form no contacts with the intermediate layer strands. Directly across the pseudo-octagonal outer layer from these two strands are two outer layer strands that each form antiparallel interactions with two intermediate layer strands. Thus, on any one cross-section, some strands have strong inter-strand contacts, while others do not contact their neighbors (asterisks in Fig. 9a-c).”

c. It would also help if a figure were to be added with a functional model of the ParMRC segregation system based on the structural data presented here and previously published.

We chose to address this point in words rather than as a figure: “The characteristics of the pCBH ParMRC elements allow for a hypothesis for the mechanism of pCBH plasmid segregation in *Clostridium botulinum*. We propose that multiple copies of the adaptor protein ParR bind to the *parC* repeats on the pCBH plasmid in a circular arrangement, as determined for the pSK41 and R1 plasmids^{21,22}. ParR from the R1 plasmid has been shown to bind to the barbed end of ParM monomers and associate with barbed ends of ParM filaments²³. The pCBH ParM filament presents different numbers of barbed end protomers at the two ends of the filament, 6 and 9, which include two circular arrangements of 6 and 8 barbed end protomers. Thus, two copies of the pCBH plasmid can be expected to bind to a

single elongating bipolar 15-stranded pCBH ParM filament, one at each end, via some of the copies of the circularly-arranged plasmid-bound ParR. The remaining copies of plasmid-bound ParR bind to ParM monomers. Elongation of the pCBH ParM filament ends at the attachment site of the two plasmids will then proceed via the ParR-bound ParM monomers joining the filament with the release of the filament-bound ParRs. In this mechanism, the plasmids remain attached to the filament and ParM polymerization provides the force to drive the plasmids to the two extremes of the *Clostridium botulinum* cell. The pointed ends of strands would elongate by incorporating free pCBH ParM to match the barbed-end strands' elongation rates, to maintain the integrity of the filament."

d. p3 "increasing levels of pCBH ParR resulted in a defined mobility shift" Please be more specific about the concentrations of pCBH ParR used (both in the text and Extended Data Figure 1).

We address this point by the sentence "Titration of DNA fragments generated via PCR from pCBH *parC* with increasing levels of pCBH ParR resulted in a defined mobility shift to larger molecular size, consistent with a specific interaction between pCBH ParR and pCBH *parC* (Fig. 1d)." and in the Fig 2 legend "EMSA of pCBH *parC* (20 nM) with increasing ratios of pCBH ParR indicated in μM ." We include the ParR concentrations in the figure.

e. What was the rationale for the design of the non-polymerizing quadruple mutant of ParM (based on what is known about the contacts in the filament?)

This is now explained in detail: "Using the *Escherichia coli* R1 ParM cryoEM protofilament structure as a template, we designed a quadruple mutant of pCBH ParM that was predicted to prevent polymerization (Supplementary Figure 3a). Mutations to subdomain 2 (F42D, I46D) and subdomain 3 (S298D and R299D) were expected to disrupt intra-protofilament subunit contacts, with subdomains 1 and 4. X-ray crystallography studies on the quadruple mutant (Supplementary Figure 3a), using the cryoEM density map (Fig. 3a) as a molecular replacement model, led to the 3.25 Å structure of the monomer (G-ParM, Fig. 3b), which was subsequently used as a guide to construct the filament protomer structure in the cryoEM density (F-ParM, Fig. 3a,c). Two of the mutation sites (Phe42 and Arg299), from the quadruple mutant, were observed to participate in intra-protofilament subunit contacts. This accounts for the success of the quadruple mutant in the preventing filament formation and allowing for monomer crystallization."

f. The abstract and text state that the ParM filament is stiffer than the actin filament. What is the evidence for this? Persistence length measurements? Or is this simply based on visual inspection by cryo-EM? This is an important point, but needs experimental support.

We have now included: "Estimation of the persistence length of the pCBH ParM filaments from the cryoEM images is 35 μm , which compares to 11 μm for the actin filament by the same method, consistent with previous reports (10-11 μm)^{12,13}. These estimations will be dependent on solution conditions, nucleotide state and the thickness of the ice, however they indicate that the pCBH ParM filaments are substantially stiffer than actin."

g. p3 "the metal ion" What is the ion?

Now stated as “magnesium ion”

These are just but a few examples of how the text and Figures could be improved. Overall, however, this is an important contribution suitable for publication in NC

Reviewer #3 (Remarks to the Author):

The manuscript by Koh et al provides an extensive study of the structure of the 15-strand parM filament from Clostridium botulinum. Despite a large number of experiments that yielded high resolution structure of the polymer the paper lacks crucial information on methods and key parameters of the pCBH parM filament. The lack of any useful discussion contributes to my pessimism regarding publishing this paper in its current form in any high profile journal.

We thank the reviewer for the constructive criticism to improve the manuscript.

I urge the authors to constructively resolve all the issues appended below:

1. The intensity of light scattering upon pCBH polymerization in Fig 1A does not match the intensities shown in SI Fig 1B. For example, the intensity obtained for 20 μ M ParM in SI Fig 1B is lower than that for the 15 μ M parM shown in Fig 1A. Therefore, it is not clear which intensities were used in the plot in Fig 1B to calculate critical concentration.

The referee is correct in pointing out the discrepancy. This is due to the experiments being carried out with different settings on the fluorimeter. All experiments under the same settings were consistent. Unfortunately, the laboratory has closed in the meantime, so currently we are not able to repeat the experiments. Given that Professor Pollard also had questions concerning this supplementary figure panel, we have removed it to avoid including incomplete data.

2. Scale bars in SI Fig 1F are unreadable!

This has been corrected, now Fig. 2b

3. There is a very poor match between the layer lines obtained from the negatively stained and frozen parM filaments in SI Fig 2. In line with that, the indexing of the layer lines is missing and thus it is not clear how the helical parameters were determined. The layer lines should be indexed and corresponding helical lattice plot should be provided.

Response: Extended Fig. 2c (now SI Fig 1) is very noisy because it comes from only one tomogram. The layer line positions in Fig. 1a and 1d match well. The impression might seem different because the signal in the helically averaged tomogram (SI Fig. 2d) was strong axially but weak radially in comparison with the averaged diffraction from negatively stained sample (SI Fig. 2a). Indexing the layer lines is very complicated and we found that it was impossible to index it by using the usual helical lattice plot reflecting the complicated structure. This is the reason that we performed cryo-electron tomography for determining the helical parameters.

4. I'm skeptical that 4.2 Å resolution can be obtained from the tomogram series collected using regular CCD camera. It is even more puzzling that from the data collected using direct detector on Titan Krios TEM the authors obtained 4.7 Å map.

On top of that, it is not clear whether all frames were used for 3D-reconstruction (with a dose of 54 e/Å²!) or dose fractioning was used.

This was badly phrased in the original manuscript. Tomography was used only for determination of helical parameters. The final structure was determined from limited frames with a dose of 30 e/Å² which is now reported in the methods. We now include: “The 2D class averages indicated a complex filament architecture (Fig. 2d), as did the averaged Fourier transform calculated from 50 negatively stained filament images (Fig. 2b). Due to this complexity, the helical parameters were determined by cryoelectron tomography (Supplementary Figure 1, molecular distance 5.2 nm, twist -50.1°). These parameters refined to a distance 5.03 nm and twist -50.4° with the cryoEM data. Helical averaging of the cryoEM density, from each cross-section of the filament, based on these parameters led to a 4.7 Å map for the entire filament (Supplementary Figure 2a,b). Within each cross-section, an intermediate layer consisting of 6 hexagonal protomers showed the best local resolution. Inter-strand averaging for this intermediate layer led to a 4.2 Å map for the protofilament (Fig. 3a and Supplementary Figure 2).” and: “Frames 3-42 of the raw movies were used for the final reconstruction, the total dose was ~ 30 electrons per Å².”

5. Regarding the reconstruction approach - the methods section does not provide any information on whether the point group symmetry was used. I assume that the helical parameters provided for “slices” correspond to a point group symmetry rotation and translation. It is also unclear how the averaged strand (SI Fig 3) was obtained taking in account that individual protofilaments forming the central, intermediate and outer layers have different helical arrangements which are not spelled out in the manuscript. It is also not clear how these symmetries are related to the helical parameters found for the parM filament earlier (Galkin et al., 2009; Bharat et al., 2015). The interface between parM protomers described in the paper should be carefully compared with the previously published data (Bharat et al., 2015).

Point group symmetry does not exist in the structure. The symmetry is just a simple one-stranded helix (-50.4 degrees and 50.3 Å). The strand-strand interactions are completely different from the previous ParMs, actin filaments or microtubules. Single strands, not pair of strands, are the structural unit. They interact with each other by flexible and non-specific electrostatic interactions. The accumulation of flexible interactions makes the rigid structure. We detail the changes in the next section that we made to the text to highlight these points.

6. The authors do not discuss how the inter-strand contacts found in the pCBH filament are related to the interactions observed between parM doublets formed in presence of PEG 6000 (Bharat et al., 2015).

We now address this point in two places: Firstly – “This architecture is completely different to the *Escherichia coli* R1 plasmid ParM filament structure, in which two protofilaments tightly associate in a staggered, parallel arrangement⁶. Furthermore, two R1 ParM filaments can come together in an antiparallel, staggered architecture⁶. None of the R1 ParM inter-protofilament interactions resemble those observed in the pCBH ParM filament.” Secondly - “The A1 loop is missing from the actin structure (Fig. 3,4 and Supplementary Figure 4) and involved in minor contacts between the two parallel strands of the ParM R1 filament⁶. The A2 loop is extended in pCBH ParM relative to actin and ParM R1 (Fig. 4). This loop lies on the outside of the actin filament¹⁶ and forms some secondary contacts between ParM R1 antiparallel filament doublets⁶. Thus, these loops are not used in assembling the

actin filament and play minor roles in assembling ParM R1 filaments and filament doublets.”

7. *The manuscript missing crucial discussion regarding how the observed structure of the pCBH filament may promote segregation of extra-large plasmids and what is the difference in the size of plasmids between different bacteria that use parM-like proteins for plasmid segregation.*

The discussion is now expanded to address this point. “The characteristics of the pCBH ParMRC elements allow for a hypothesis for the mechanism of pCBH plasmid segregation in *Clostridium botulinum*. We propose that multiple copies of the adaptor protein ParR bind to the *parC* repeats on the pCBH plasmid in a circular arrangement, as determined for the pSK41 and R1 plasmids^{21,22}. ParR from the R1 plasmid has been shown to bind to the barbed end of ParM monomers and associate with barbed ends of ParM filaments²³. The pCBH ParM filament presents different numbers of barbed end protomers at the two ends of the filament, 6 and 9, which include two circular arrangements of 6 and 8 barbed end protomers. Thus, two copies of the pCBH plasmid can be expected to bind to a single elongating bipolar 15-stranded pCBH ParM filament, one at each end, via some of the copies of the circularly-arranged plasmid-bound ParR. The remaining copies of plasmid-bound ParR bind to ParM monomers. Elongation of the pCBH ParM filament ends at the attachment site of the two plasmids will then proceed via the ParR-bound ParM monomers joining the filament with the release of the filament-bound ParRs. In this mechanism, the plasmids remain attached to the filament and ParM polymerization provides the force to drive the plasmids to the two extremes of the *Clostridium botulinum* cell. The pointed ends of strands would elongate by incorporating free pCBH ParM to match the barbed-end strands’ elongation rates, to maintain the integrity of the filament. The pCBH ParM filament appears to be highly adapted to segregating two large plasmids, which are 245-267 kb in size for plasmids containing this ParMRC system. In comparison, the *Escherichia coli* R1 plasmid is 98 kb in size and is segregated by the antiparallel association of two 2-stranded ParM filaments^{6,24}. The pCBH ParM “actin-like” filament has a similar number of protofilaments and diameter (26 nm) to chromosome-segregating microtubules¹. The pCBH ParM filament is bipolar and stiff, and like microtubules, the 15 polymerizing strands are likely to exert greater combined force relative to typical 2-stranded actin-like filaments.”

REVIEWERS' COMMENTS:

Reviewer #3 (Remarks to the Author):

In the current form the manuscript is acceptable for publication, but I still would like the authors to do minor modifications appended below:

1. P2 L38: Replace "more complex actin-like filament formed..." with "more complex filament formed by the actin-like protein associated into 15 protofilaments". The filament comprised of 15 strands cannot be an actin-like filament.
2. P2 L44: Replace "to produce a filament diameter ..." with "to produce a filament with a diameter of...".
3. P4 L107: I would suggest to use "axial rise" instead of "molecular distance" when related to helical symmetry parameters
4. In Fig 3a only α -helices are shown. At the claimed resolution bulky side chains should be visible. I would like the authors to add close-up inserts to demonstrate visibility of large side chains within the electron density map.
5. The mechanism of plasmid segregation is hard to understand without a figure, therefore, I would like the authors to make one more figure to illustrate polymerization/segregation mechanism proposed in the final paragraph.

REVIEWERS' COMMENTS:

Reviewer #3 (Remarks to the Author):

In the current form the manuscript is acceptable for publication, but I still would like the authors to do minor modifications appended below:

1. P2 L38: Replace "more complex actin-like filament formed..." with "more complex filament formed by the actin-like protein associated into 15 protofilaments". The filament comprised of 15 strands cannot be an actin-like filament. We have reworded to "a complex filament formed from 15 protofilaments of an actin-like protein."

2. P2 L44: Replace "to produce a filament diameter ..." with "to produce a filament with a diameter of...". We have reworded to "In cross-section, the ~26 nm diameter filament comprises..."

3. P4 L107: I would suggest to use "axial rise" instead of "molecular distance" when related to helical symmetry parameters
Changed as suggested.

4. In Fig 3a only c-alphas are shown. At the claimed resolution bulky side chains should be visible. I would like the authors to add close-up inserts to demonstrate visibility of large side chains within the electron density map.
The side chains are now included in Supplementary Fig. 3b.

5. The mechanism of plasmid segregation is hard to understand without a figure, therefore, I would like the authors to make one more figure to illustrate polymerization/segregation mechanism proposed in the final paragraph.
The figure is now included - Figure 10.